# Reward network mechanism in anhedonia and depression

**Xiaoxiao Sun**[1☺], **Chenxuan Jin**[1,2☺], **Chun Cui**[3], **Zhaohua Chen**[1], **Qin Dai**[1,4*]

**1** Department of pain, the Southwest hospital, Army Medical University, Chong qing, China, **2** Joint Logistics Support Force Hospital 958, People's Liberation Army, China **3** Radiological Department, the Xinqiao Hospital of the Army Medical University, Chong Qing, China, **4** College of General Education and International, Chongqing Polytechnic University of Electronic Technology, Chong qing, China

☺ The authors contributed equally to this work.

* daiqin101@hotmail.com

## Abstract

### Background

Depression is one of the most burdensome mental disorders. Anhedonia, a core symptom of major depressive disorder (MDD), is characterized by abnormal resting-state reward network (RN). However, it is unclear whether anhedonia symptom and depressive episode share similar resting-state RN mechanism, as well as whether the RN mechanism is a state or trait-like marker of depression. This study aims to clarify the two points by recruitingboth current and remitted depression.

### Methods

Using functional Magnetic Resonance Imaging (fMRI) scans, this study observed the resting-state RN function connectivity (with the seed of ventral striatum) in patients with remitted depression (RMD, n = 27) and current depression (n = 30) and 33 normal controls. The low-frequency fluctuation (ALFF) and T1 image were further analyzed.

### Results

Three groups differed in anhedonia scores, with highest anhedonia in the MDD group and lowest anhedonia in the NC group. In total sample, higher anhedonia was correlated with weaker connectivity between the striatum seed and the putamen, inferior frontal cortex, insula, AC, and thalamus, while in the RMD group, anhedonia correlated with higher AC, thalamus, and caudate connectivity. In resting-state function connectivity, the MDD group possessed weaker connectivity between the striatum seed and inferior frontal cortex and insula, while the RMD group showed weaker connectivity with the caudate, and both the MDD and RMD groups possessed lower connectivity with the AC. ALFF data indicated a higher anterior cingulate (AC) activation

**Data availability statement:** The behavioral data that support the findings of this study were included in Supplementary materials. Due to reframing from military funding (2021HL003 and 2023yjsB02), the neural data could not be placed publicly and was available from the corresponding author upon reasonable request. To ensure the long term stability and availability of data, the neural data was packed and stored in three copies by a non-author institution (Military Psychology Committee institute, Chinese Society of Social Psychology) to ensure persistent or long-term data storage and availability, which was available from the official email address of this committee (jsxlx-zwh@163.com).

**Funding:** Doctor Dai claimed that this work was supported by Social Science Foundation of Chongqing (2023NDYB94), the Key project and innovation project of People's Liberation Army of China (2021HL003), and the education project of Army medical university (2023yjsB02). The funders had no role in study design, data collection and analysis, decision to publish, or preparation of the manuscript, only reframed the sharing of original neural data.

**Competing interests:** Authors claimed receiving research funding from Social Science Foundation of Chongqing (2023NDYB94), the Key project and innovation project of People's Liberation Army of China (2021HL003), and the education project of Army medical university (2023yjsB02). This does not alter our adherence to PLOS ONE policies on sharing behavioral data, only limits on original neural data sharing from military funding (2021HL003 and 2023yjsB02). Authors reported no biomedical financial interests or potential conflicts of interest. All authors stated that the content has not been published or submitted for publication elsewhere.

in the MDD group than the RMD group.T1 image indicated a bigger thalamus volume in the MDD group than the RMD group.

## Conclusions

The current study is among the first to confirm that RMD patients possess different RN pattern compared with MDD. Importantly, caudate playsa unique role in depression remission, AC and thalamus mechanisms are trait-like markers of depression. Surprisingly, insula and inferior frontal mechanisms share by depressive episode and anhedonia, while putamen discriminates depressive episode and anhedonia. The results suggest candidate biomarkers for the treatment of clinical depression.

## Introduction

Depression is one of the most burdensome mental disorders [1], which has not been well prevented (3% to 22.5% incidence) [2] or cured (one-third non-responder and high recurrence:60% after 5 years, 67% after 10 years, and 85% after 15 years) [3] despite decades of efforts. To better control depression, increasing studies have assessedthe neural alterations of depression in an attempt to reveal potential cerebral mechanisms underlying the occurrence and recurrence of major depressive disorder (MDD) [4]. Neuro-imaging findings have identified that MDD patients showed decreased reward network (RN) connectivity (mainly in the prefrontal-striatal regions) [5,6], which comprised of ventral and dorsal striatum, medial prefrontal-cortex (including orbito-frontal cortex), anterior cingulate, medial temporal lobes, and ventral tegmental area [7]. Blunted left ventral striatum response to reward was associated with a lifetime depression diagnosis [8]. Recent review indicated that compared with healthy controls, patients with MDD exhibited common activity decreases in the right striatum (putamen, caudate) and subgenual ACC [9]. Functional connectivity between the ventral striatum (key region of the RN) and the default mode network was found to be positively and significantly associated with depression scores [10]. Moreover, RN connectivity was found to be correlated with antidepressant therapy in which changes in the sensitivity of the neural reward were positively correlated with the improvement of depressive symptoms [11]. Consistently, MDD patients who do not respond to repetitive transcranial magnetic stimulation therapy were found to show significantly weaker connectivity in the traditional reward network [12]. These results suggest a role of the neural RN in depressive episode.

A core symptom of major depressive disorder is anhedonia, which has beenoften defined as hyposensitivity to reward (this perspective focuses on emotional arousal to reward) [13], and also defined sometimes as diminished anticipation and pursuit of reward (this perspective focuses on motivation and behaviorresponses to reward) [14]. The cerebral foundation of anhedonia has been explored. In which, the hypo-arousalcould be well observed in resting-state neural images [ 12, 15], while the motivation and behavior responses could be observed by money-reward task [16]. Specifically, it was found that anhedonia was correlated with non-response to

transcranial magnetic stimulation which had lower resting-state connectivity in the classical reward pathway [12]. Consistently, a disturbance in resting-state reward network has been confirmed to be correlated with depressive anhedoniain rats [15]. Change in anhedonia scores negatively correlated with rsFC after antidepressant treatment [17]. A previous study further reported that anhedonia symptom and depressive mood were associated with similar striatal circuits in MDD patients with money-reward task [16]. Indeed, RN was indicated associated with both anhedonia and depression [18]. The results indicated a role of RN, especially the ventral striatum, in anhedonia. However, anhedonia is a core symptom of depressive episode, anhedonia itself can not represent a depressive episode [19]. Specifically, to be diagnosed with MDD, one must be in depressed mood or anhedonia which causes social/occupational impairment [20], i.e., anhedonia is only one symptom of MDD which might be exempted from experiencing by some patients with MDD [19]. If some patients experience anhedonia while others not, it thus could be deduced that the neural mechanism underling MDD may be differed but alike with anhedonia (**Hypothesis 1**), since that MDD was formed by broader symptoms such as sadness or insomnia which were very different from anhedonia [21,22]. However, further evidenceis needed to confirm whether anhedoniasymptom and depressive episode share similarbutsubtly-differentresting-state reward network mechanism, clarifying this issue might be important to suggest candidate cortexfor clinical depression therapy, especially for anhedonia-type depression.

Importantly, depression is a highly recurrent mood disorder [23]. Comparison between current and remitted depression may suggest trait- or state- marker of depression, i.e., commonality between them suggests a trait-like marker while difference between them suggests a state-like marker [24–26]. However, we do not know whether the RN mechanism is a state marker or a trait-like character closely correlated with the onset or recurrence of depression. Unfortunately, although a few studies observed neural differences between current and remitted depression [27], most studies explored the neural activation of current and remitted depression separately [5,12,28–30]. Among these studies, current depression was observed most often [5,12,30], while few studies focused on remitted depression [28]. A limited number of researches reported that remitted depression hasa different RN activation during a reward process (hyperactivation while anticipating a reward in bilateral anterior cingulate gyrus and right midfrontal gyrus and hypoactivation during reward outcomesin orbital frontal cortex, right frontal pole, left insular cortex, and left thalamus) [28]. In contrast, in another study, hypoactivation in the putamen and caudate of a remitted patient was observed [29]. It is well known that the current and remitted depression differin symptoms [31] and neural foundation [25,26], it could be assumed that they differ significantly in resting-state RN mechanism (**Hypothesis 2**). Therefore, this study included patients with remitted depression (RMD) to confirm the role of RN mechanism during the remission of depression, and further compare with that of MDD to confirm whether RN mechanism is a state marker of depressive episode or a trait-like mechanism related to the occurrence and recurrence of depression.

Thus, the present study had two aims. The first was to explore thedifferencesinthe RN mechanismsbetween anhedonia and depressiveepisode. The second was to explore the differencesinthe RN mechanismbetween current and remitted depression. Our hypotheses wereas follows: (1) Anhedonia and depressive episodemay share similar but subtly-different RN mechanism. (2) Current and remitted depression may share different RN mechanisms.

## Methods

### Participants

To compare the neural network connectivityin individuals with different depressive statuses, we recruited never depressed normal controls (NCs) and patients with remitted and current depression from April of 2021 to January of 2022. The depressed patients were recruited throughthe psychiatrists inclinic, and the healthy controls were recruited via an advertisement. The depressive symptoms were evaluated using the Beck Depression Inventory-II (BDI-II) [32,33], the Patient Health Questionnaire (PHQ-9) [34,35], the Structured Clinical Interview for Diagnostic and Statistical Manual for Mental Disorders-5th (SCID) [36] and the Hamilton Depression Rating Scale (HDRS, 24 items) [37]. To investigate the anxiety status of the

participants, the Beck Anxiety Inventory-II (BAI) was also used. The inclusion criteria for the three groups were as follows: the normal controls should not have a lifetime or current psychosis or depression, the MDD patients should be diagnosed as having major depression by a clinical psychiatrist and have had two or more episodes according to the symptoms of the DSM-5[th] [38],and the RMD individuals should be antidepressant free for 3 months or longer and recovered from their last episode for 6 months or longer (the mean remission time was 9.24 months). The exclusion criteria were as follows: current or history of any psychopathology and/or drug and alcohol abuseinfection, drug dependence, allergies within the last half month, physical trauma, severe physical disease, learning or cognitive disability. Due to the potential risk from a fMRI scan, subjects with the following situations were also excluded: being pregnant or having a pacemaker, any magnetic or metal materials within the whole body, or having epilepsy or a history of brain surgery. The recruitment details were shown in our previous work and in S1 Fig [39]. Thirty MDD patients, twenty-seven remitted individuals, and thirty-three normal controls were recruited. All MDD patients took medicine (eight were taking fluoxetine hydrochloride, 20–40 mg; ten were taking mirtazapine, 30–40 mg; and twelve were taking paroxetine hydrochloride, 20–30 mg). See S1 Fig for the details.

## Questionnaire

The PHQ-9 and BDI-II were utilized to evaluate the levels of depression. The PHQ-9 assessed the symptoms of depressioncorresponding to the frequency of 9 diagnostic items ofmajor depression, and the BDI-IIexamined the severity of depression with 21items [32,40]. The first item (frequency: loss of interest and pleasure) of PHQ and the fourth item (severity: loss of pleasure) of BDI-II were further used to evaluate the anhedonia symptoms of subjects. The BAI also examinedthe level of anxiety with 21items [40].

## Procedures

All procedures performed in studies involving human participants were in accordance with the ethical standards of the institutional and/or national research committee and with the 1964 Helsinki Declaration and its later amendments or comparable ethical standards. The study protocol including participants'capacity to provide consentin accordance with the Declaration of Helsinki was approved by the Ethics Committee of Human Research of the Army Medical University (2020-036-02) and Chinese Clinical Trial registry center (ChiCTR2100044258). The clinical patients with MDD were screened and recruited by the psychiatrists in clinicbased on the SCID and the HDRS. The potential subjects (patients and normal controls) were screened for recruitment andtheircapacity to provide consentby trained personnel withdoctoral degreewho major in psychology via telephone, and qualified participants were scheduled for appointments after the interview via telephone. After the subjects arrived, they were orally informed about the detailed procedure of this study and were further informed about their right to quit any time during the study without any negative outcome. After the subjectssigned on the written informed consent forms, a 10-minute rest period was allowed before the formal study to help subjects to become familiar with the study environment.After the study, participants were available with free psychological consultation and gifted (50 Yuan for each participant).

## MRI image acquisition

The brain images were obtained from a whole-body MRI system (Germany, Erlangen,3.0-T Siemens TimTrio) with a 12-channel phased-array head coil, located at Southwest Hospital of Chongqing, China. The image data were scanned via single shot T2-weighted EPI (echo planar imaging, TR/TE = 2000/30 ms, reconstruction matrix = 64 × 64, flip angle = 90°, FOV = 384 mm, voxel size = 3.0 × 3.0 × 3.0 mm³, 4 mm slice thickness with 1 mm gap, and number of slices foreach volume = 36). The duration of the scan for the resting state was 8 minutes (240 volumes). The anatomical images were obtained viathe 3D-T1MPRAGE protocol (TI/TR/TE = 900/2530/2.34 ms, reconstruction matrix = 256 × 256, flip angle = 7°, slice thickness = 1 mm, FOV = 256 mm, and number of slices = 192).

## MRI image analyses

The cerebral activation images were processedusingSPM8.

**Preprocessing** The first 10 scan images were removed. Then, the remaining images were preprocessed via SPM8. The brief procedures included the following: (1) slice timing, (2) realignment, (3) coregistration, (4) segmentation, (5) normalization (via DARTEL with a re-sampling rate of $3 \times 3 \times 3mm^3$ ($1 \times 1 \times 1mm^3$ for anatomic images)), and (6) smoothing (size of kernel smoothing = 6 mm (FWHM)). The head motion, cerebrospinal fluid, and white matter were regressed out during the preprocessing with DPARSFA [41].

## Resting-state data

REST 1.8 was used to analyze the amplitude of low-frequency fluctuation (ALFF) of resting state [41]. In general, linear regression was conducted on data after preprocessing to remove linear tendency. Hamming band-pass filter was then carried out to remain thesignals between 0.01 and 0.08 Hz (low-frequency). The ALFF value of whole-brain voxel was computed, which was used to divide the average ALFF, and resulted in a standard ALFF value [42].

The functional connectivity between cerebral cortexes was analyzed using REST 1.8 (Resting-State fMRI Data Analysis Toolkit) [43], andautomated anatomical labeling (AAL) was used [44]. The MNI coordinates of the ventral striatum (21, 9, 0) were consistent with the literature [45]. RN connectivity was obtained from the whole-brainfunctional connectivity with this ventral striatum seed. See the supplementary materials (S2 Fig) for details.

## T1 image

T1 image was analyzed with SPM8 via DARTEL (voxel-based morphometry, VBM) [44]. The brief procedures included the following: (1) segmentation; (2) normalization; (3) nonlinear registration; (4) coregistration; (5) smoothing. One-way ANOVA was conducted on T1 image data between the three groups.

## Statistics

One-way ANOVA was conductedon T1 and ALFF image data andthe functional connectivity between the three groups. A subsequent independent t-test was conducted in each of the two group comparisons (AlphaSim correction, corrected pvalue < .01 (uncorrected p value < .001), in a brain volume of $61 \times 73 \times 61$, estimated spatial smoothness of 6 mm, and a minimum cluster size of 26 voxels or $702 mm^3$ (AFNI; https://afni.nimh.nih.gov/pub/dist/doc/manual/AlphaSim.pdf), RN mask (S2 Fig)). Moreover, a correlation analysis was also conducted between the value of resting-state RN connectivityand anhedonia scoresusing REST 1.8 [43]. The analyzed images were configured using Rest Viewer (overlaid above a ch2 bet.nii template). Age and gender were controlled as covariables.

The demographic information and scores of scales of the participants were compared via one-way ANOVA, while the ratio of gender and anhedonia were evaluated by $\chi^2$ test.

## Results

### General information (Table 1)

Three groups did not significantly differ in age (F (2,87) = 0.038, partial $eta^2$ = 0.001, p = 0.962), education level (F (2,87) = 0.112, partial $eta^2$ = 0.003, p = 0.894), or male/femaleratio ($\chi^2$ = 2.001, p = 0.368, df = 2, N = 90). As hypothesized, they significantly differed on theBDI-II (F (2,87) = 223.166, partial $eta^2$ = 0.84, p < 0.001), PHQ (F (2,87) = 252.477, partial $eta^2$ = 0.853, p < 0.001), HDRS (F (2,87) = 624.037, partial $eta^2$ = 0.94, p < 0.001), and BAI scores (F (2,87) = 58.101, partial $eta^2$ = 0.572, p < 0.001), with the lowest scores in the NC group and the highest scores in the MDD group. The remitted and current depression groups did not significantly differ in the duration of depression or the number of episodes (p > 0.05).The three groups did not significantly differ in headmotion (F (2,87) = 2.137, partial $eta^2$ = 0.047, p = 0.124).

## Table 1. General information.

| | Gender (M/F) | Age | Education (years) | Episodes | Duration of illness (years) | FD value of head motion | Score of PHQ | Score of BDI | Score of HDRS | Score of BAI |
|---|---|---|---|---|---|---|---|---|---|---|
| NC (n = 33) | 11/22 | 48.85 ± 12.38 | 12.33 ± 2.33 | | | 0.13 ± 0.05 | 1.12 ± 1.24 | 2.21 ± 1.45 | 1.92 ± 1.24 | 1.77 ± 1.36 |
| MDD (n = 30) | 7/23 | 48.77 ± 11.88 | 12.47 ± 2.69 | 2.80 ± 1.35 | 9.20 ± 1.94 | 0.12 ± 0.06 | 21.63 ± 5.25 | 34.23 ± 10.08 | 23.27 ± 2.95 | 12.97 ± 6.75 |
| RMD (n = 27) | 11/16 | 49.56 ± 10.77 | 12.11 ± 3.52 | 2.63 ± 1.33 | 8.67 ± 2.66 | 0.15 ± 0.06 | 5.41 ± 3.87 | 6.74 ± 4.59 | 5.67 ± 3.11 | 4.63 ± 2.63 |
| P (ANOVA/ $\chi^2$ test) | 0.368 | 0.962 | 0.894 | 0.634 | 0.387 | 0.124 | **<0.001** | **<0.001** | **<0.001** | **<0.001** |

Note: M = male. F = female. BAI = Beck Anxiety Inventory. BDI-II = Beck Depression Inventory-II. PHQ = patient health questionnaire. HDRS = the Hamilton Depression Rating Scale. NC = Never disordered healthy Controls. MDD = Major Depressive Disorder. RMD = Remitted depression.

### Anhedoniasymptoms of the participants (Table 2)

Due to the close relationship between anhedonia and depressive episode, ANOVA was conducted on the anhedonia items of the questionnaires to compare the anhedonia symptom between groups. The results confirmed the expected differences between three groups on the scores of anhedonia item (first item) of the PHQ (F (2,87) = 99.031, p < 0.001, partial eta$^2$ = 0.70) and the fourth item of the BDI-II (F (2,87) = 106.399, p < 0.001, partial eta$^2$ = 0.71), with the lowest scores in the normal controls and highest scores in the MDD group. Consistently, more individuals in the MDD group (all except one) reported suffering from anhedonia compared with the other two groups, based on the scores of first item of PHQ ($\chi^2$ = 42.106, df = 2, N = 90, p < 0.001) and fourth item of BDI-II ($\chi^2$ = 60.705, df = 2, N = 90, p < 0.001).

### Correlation between functional connectivity and anhedonia (Table 3)

To observe the relationship between neural function connectivity and depressive anhedonia, correlations between the value of resting-state RN connectivity and the anhedonia scores in all participants and sub-groups were conducted using REST 1.8. In total sample, a negative correlation was found between the severity of anhedonia (4th item of BDI-II, Fig 1) and striatum functional connectivity in the right putamen (r = −0.39, k = 14, p < 0.05), left inferior frontal cortex (r = −0.31, k = 21, p < 0.05), right inferior frontal cortex (r = −0.38, k = 47, p < 0.05), left AC (r = −0.41, k = 57, p < 0.05), and right insula (r = −0.43, k = 43, p < 0.01); in addition, a negative correlation was found between the frequency of anhedonia (1st item of PHQ, Fig 2) and striatum functional connectivity in the right insula (r = −0.42, k = 24, p < 0.05), left inferior frontal cortex (r = −0.33, k = 21, p < 0.05), left putamen (r = −0.29, k = 24, p < 0.05), and right thalamus (r = −0.34, k = 27, p < 0.05). In the RMD group, a positive correlation was found between the severity of anhedonia (Fig 3) and striatum functional connectivity in the right thalamus (r = 0.78, k = 50, p < 0.05); a negative correlation was found between the severity of anhedonia and striatum functional connectivity in the right caudate

## Table 2. Anhedonia symptom of participants.

| | Scores of anhedonia item | | Ratio of anhedonia (present/absent) | |
|---|---|---|---|---|
| | PHQ-1st | BDI-4th | PHQ-1st | BDI-4th |
| NC (n = 33) | 0.15 ± 0.57 | 0.03 ± 0.17 | 5/28 | 1/32 |
| MDD (n = 30) | 2.57 ± 0.77 | 2.07 ± 0.83 | 29/1 | 30/0 |
| RMD (n = 27) | 0.67 ± 0.78 | 0.41 ± 0.57 | 13/14 | 10/17 |
| F/$\chi^2$ | 99.031** | 106.399** | 42.106** | 60.705** |

Note: ** p < 0.01.

**Table 3. Correlation between RN function connectivity and anhedonia.**

| Cortex | L/R | Cluster (k) | r | MNI | | |
|---|---|---|---|---|---|---|
| | | | | x | y | z |
| **BDI4-all subjects** | | | | | | |
| Putamen | R | 14 | −0.37** | 15 | 6 | −6 |
| Inferior frontal cortex | L | 21 | −0.30** | −39 | 24 | 6 |
| Inferior frontal cortex | R | 47 | −0.29** | 54 | 24 | 3 |
| AC | L | 57 | −0.36** | −12 | 36 | 0 |
| Insula | R | 43 | −0.39** | 36 | 9 | 9 |
| **PHQ1-all subjects** | | | | | | |
| Thalamus | R | 27 | −0.26* | 9 | −9 | 3 |
| Putamen | L | 24 | −0.22* | −30 | 6 | −3 |
| Insula | R | 24 | −0.35** | 36 | 9 | 12 |
| Inferior frontal cortex | L | 21 | −0.30** | −39 | 18 | 9 |
| **RMD- BDI4** | | | | | | |
| Caudate | R | 12 | −0.57** | 6 | 9 | −3 |
| Thalamus | R | 50 | 0.52** | 6 | −12 | 3 |
| Suprior Frontal Gyrus | R | 46 | −0.70** | 9 | 12 | 66 |
| **RMD- PHQ1** | | | | | | |
| Caudate | R | 38 | 0.53** | 12 | 12 | 6 |
| AC | R | 50 | 0.47* | 3 | 33 | 21 |
| Thalamus | R | 23 | 0.55** | 18 | −12 | 9 |

Note: L: left, R: right, * p < 0.05, ** p < 0.01. MDD = Major Depressive Disorder. RMD = Remitted depression. BDI4 = Anhedonia item of Beck depression inventory. PHQ1 = Anhedonia item of patient health questionnaire.

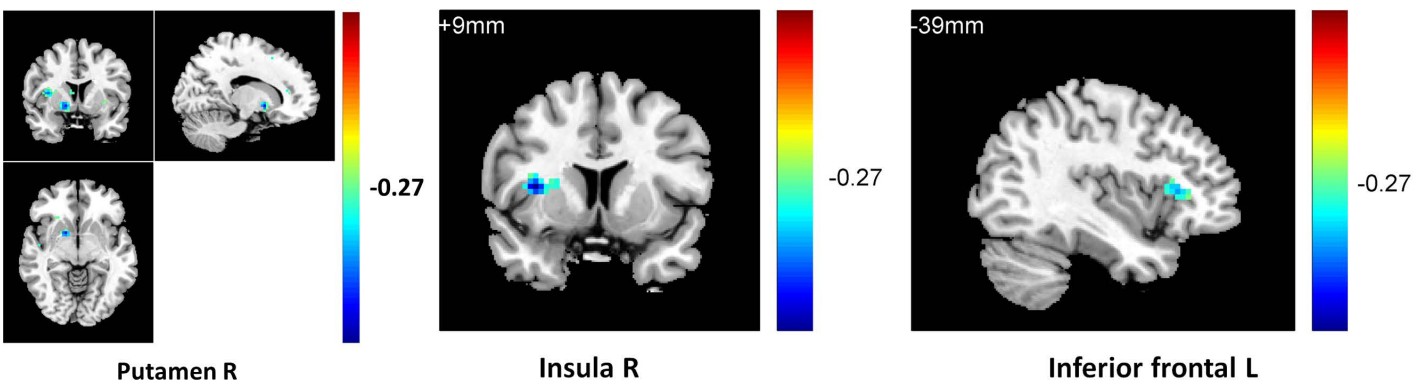

**Putamen R**   **Insula R**   **Inferior frontal L**

**Fig 1. Negative correlation between insula, inferior frontal, and putamen connectivity and depressive anhedonia (BDI4).**

(r = −0.65, k = 12, p < 0.05) and right superior frontal gyrus(r = −0.81, k = 46, p < 0.01); and a positive correlation was found between the frequency of anhedonia (Fig 4) and striatum functional connectivity in the right caudate (r = 0.77, k = 38, p < 0.01), right AC (r = 0.79, k = 50, p < 0.01), and right thalamus (r = 0.77, k = 23, p < 0.01). No significant correlation between anhedonia and striatum connectivity was foundin the NC and MDD groups. See Figs 1–2,Table 3, and S3 Fig (scatterplots) for details.

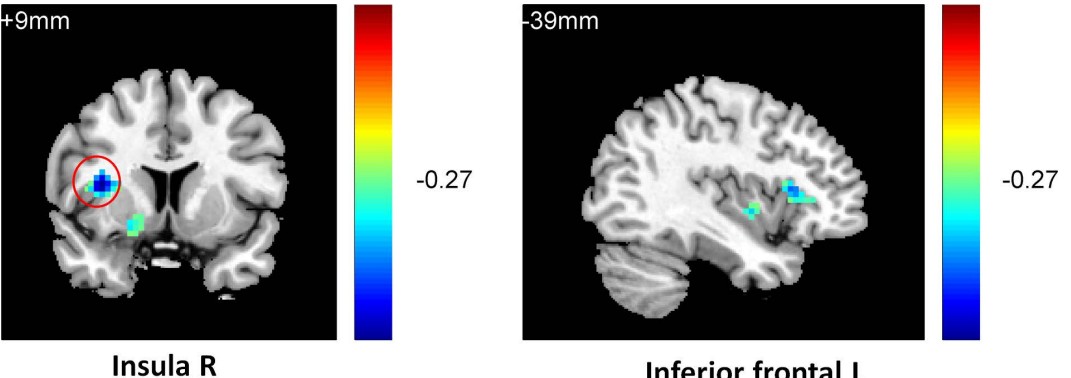

**Fig 2. Negative correlation between insula, inferior frontal, and putamen connectivity and depressive anhedonia (PHQ1).**

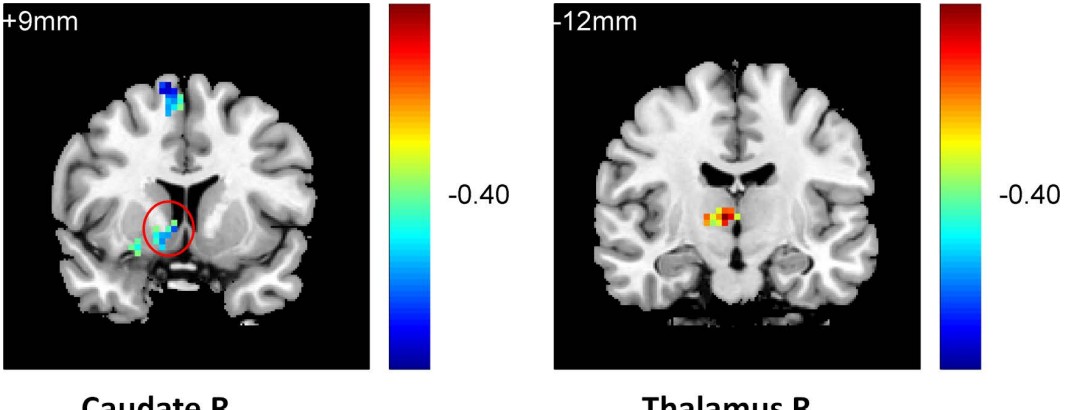

**Fig 3. Negative correlation between caudate connectivity and positive correlation between thalamus and ACconnectivityand depressive anhedonia in RMD group (BDI4).**

### Resting-state activity and function connectivity in current and remitted depression

**Functional connectivity (Table 4).** One-way ANOVA (analyzed using REST 1.8) on the functional connectivity between the striatum seed and RN indicated that the three groups differed in their connectivity between the striatum seed and the left caudate (k = 21, p < 0.01), the left anterior (k = 43, p < 0.05), and the middle cingulate gyrus (k = 24, p < 0.05), S4 Fig. Further two-sample t-tests (AlphaSim correction, corrected p-value < 0.01) indicated that compared with the normal controls, the patients with current depression exhibited weaker connectivity between the striatum seed and the right insula (k = 27, p < 0.01), the right inferior frontal cortex (k = 130, p < 0.05), and the left anterior cingulated gyrus (AC) (k = 119, p < 0.05) (Fig 5). In contrast, compared with the MDD group, individuals with remitted depression showed stronger connectivity between the striatum seed and the right inferior frontal cortex (k = 55, p < 0.05), the right insula (k = 20, p < 0.05), and the right middle frontal gyrus (k = 43, p < 0.05) and weaker connectivity between the striatum seed and the left and right caudate (k = 27, k = 26, p < 0.05) (Fig 6). They also showed weaker connectivity between the striatum seed and the left caudate (k = 117, p < 0.01) and the cingulated gyrus (k = 76, p < 0.01) compared with the normal controls (Fig 7).

**Amplitude of low-frequency fluctuation (ALFF).** An ANOVA was conducted on the ALFF data, which indicated a difference between three groups in the right ACC (k = 13, F (2,87) = 7.04, x = 9 y = 21 z = −9, p < .01). A further two-sample t

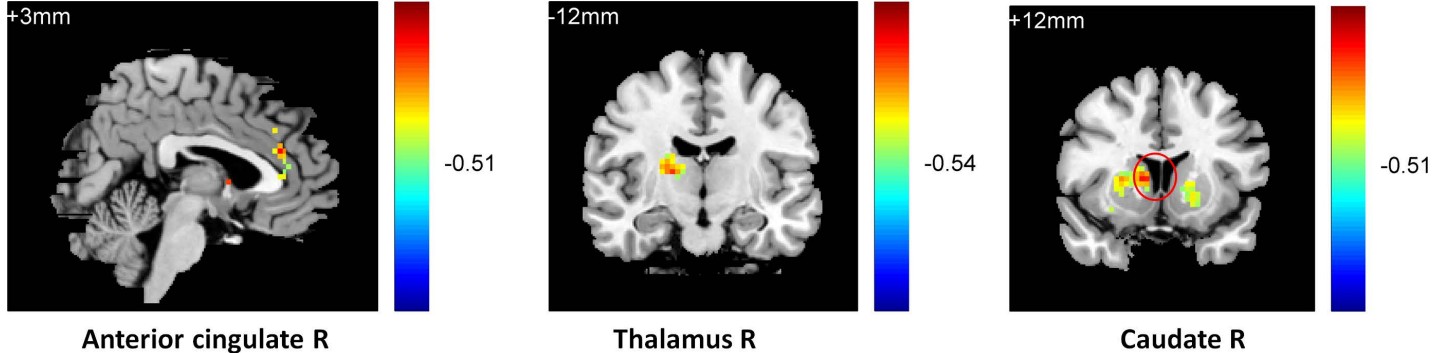

**Fig 4. Negative correlation between caudate connectivity and positive correlation between thalamus and ACconnectivityand depressive anhedonia in RMD group (PHQ1).**

**Table 4. RN function connectivity in current and remitted depression (striatum seed).**

| Cortex | L/R | Cluster (k) | | MNI | | |
|---|---|---|---|---|---|---|
| **Three groups** | | | F | x | y | z |
| Caudate | L | 11 | 14.89** | −18 | −3 | −21 |
| Anterior Cingulate | L | 43 | 6.45* | −12 | 33 | 6 |
| Middle Cingulate | L | 14 | 15.39* | −3 | −3 | 33 |
| **MDD-NC** | | | t | x | y | z |
| Insular | R | 27 | −4.40** | 36 | 9 | 9 |
| Inferior frontal cortex | R | 130 | −3.49* | 54 | 21 | 24 |
| Anterior Cingulate | L | 119 | −4.61* | −12 | 39 | 0 |
| **RMD-MDD** | | | t | x | y | z |
| Inferior frontal cortex | R | 55 | 3.83* | 57 | 21 | 15 |
| Caudate | R | 26 | −3.78* | 6 | −9 | 15 |
| Caudate | L | 27 | −3.54* | −18 | −3 | 21 |
| Insula | R | 10 | 3.41* | 21 | 12 | −9 |
| Middle Frontal Gyrus | R | 43 | 3.96* | 36 | 33 | 30 |
| **RMD-NC** | | | t | x | y | z |
| Caudate | L | 117 | −6.03** | −18 | −3 | 21 |
| Cingulate gyrus | R/L | 76 | −5.17** | −3 | −3 | 33 |

Note: L: left, R: right, * p < 0.05, * p < 0.05, ** p < 0.01.

test (AlphaSim correction, corrected p value < .01) showed that the RMD group had a lower activity inthe right ACC (k = 17, t = 3.09, x = 9 y = 21 z = −6, p < .01) compared with the MDD group.

**T1 image.** An ANOVA was conducted on the T1image, which indicated a difference between the three groups in the right thalamus (k = 40, x = 9 y = −24 z = 15, F (2,87) = 7.19, p < .01). A further two-sample t test (AlphaSim correction, corrected p value < .01) showed that the RMD group had a smaller volume inthe right thalamus (k = 79, x = 12 y = −24 z = 18,t = 3.64, p < .01) compared with MDD group. See S5 Fig for details.

## Discussions

The current study found that RMD patients possess different RN pattern of anhedonia. Potentially and importantly, caudate played a unique role in depression remission, AC and thalamus mechanisms were trait-like markers of depression.

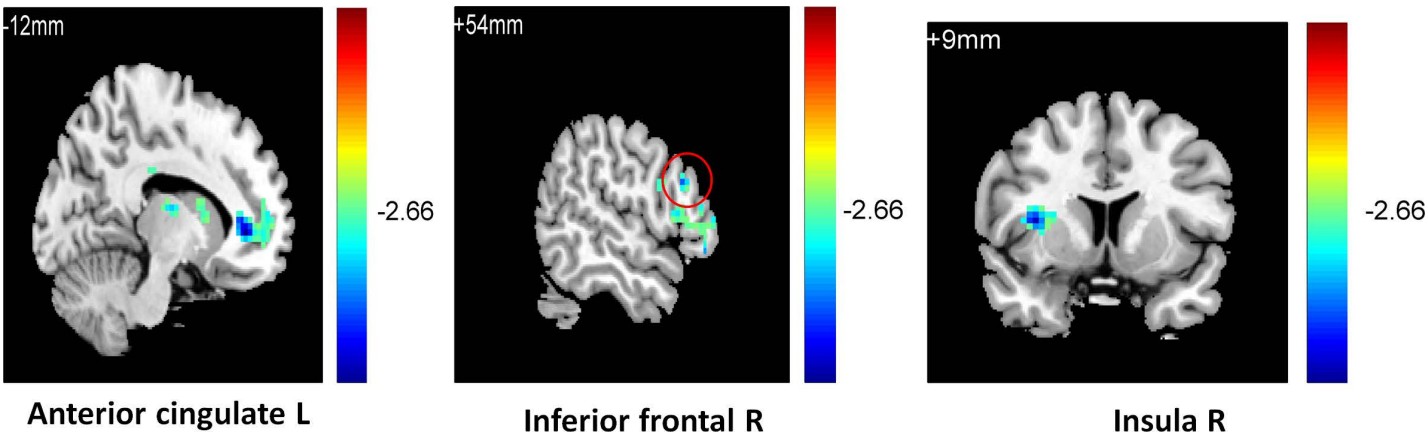

**Fig 5. Lower inferior frontal, anterior cingulated and insula connectivity in the MDD group compared with the NC group.**

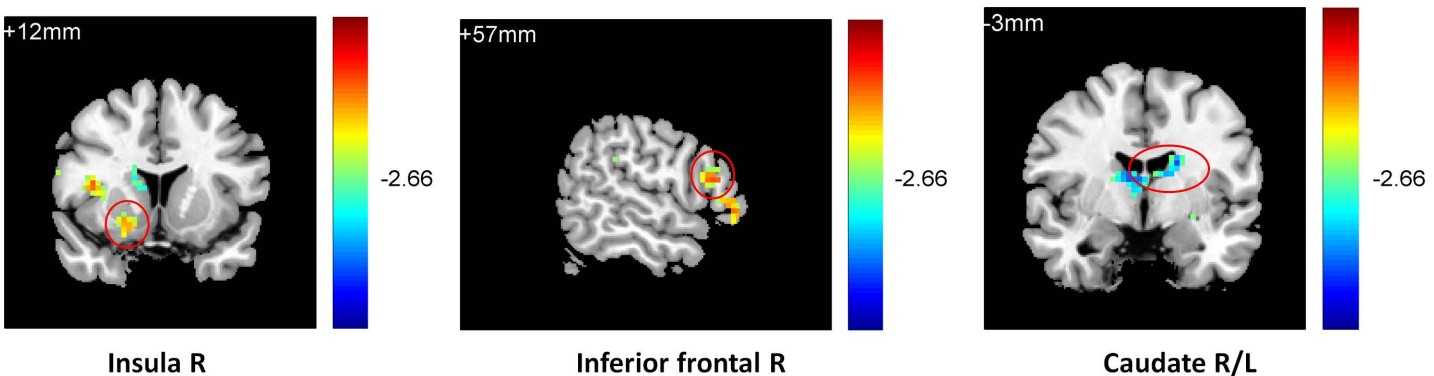

**Fig 6. Higher inferior frontal and insula connectivity and lower caudate connectivity in the RMD group compared with the MDD group.**

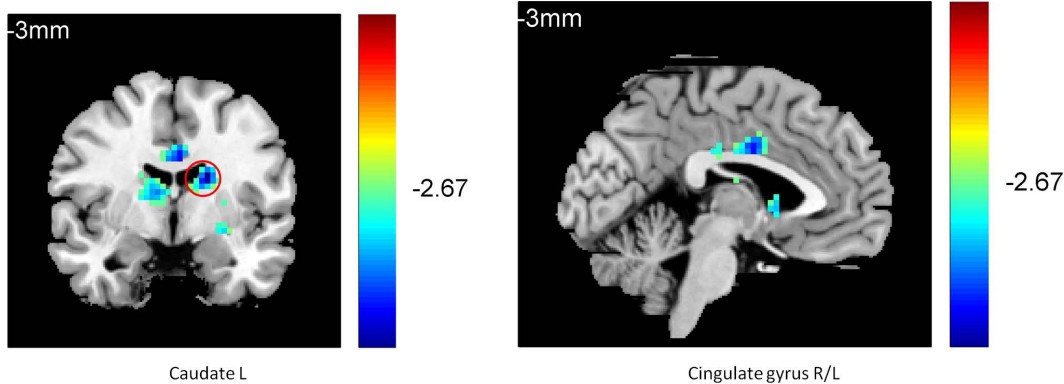

**Fig 7. Lower caudate and cingulate connectivity in the RMD group compared with theNC group.**

Surprisingly, insula and inferior frontal mechanisms shared by depressive episode and anhedonia, while putamen discriminated depressive episode and anhedonia.

Our result found that three groups differed in anhedonia scores, with highest anhedonia in the MDD group andlowest anhedoniain the NC group. This finding support that anhedonia is a core symptom of clinical depression [13], which is significantly differed in different depressive status.This study reports and compares anhedonia symptoms of current and remitted depression in one single study, which enriches the understanding about the anhedonia during the process of depression. Potentially, the differences in resting-state RN mechanism between groups may indicate the neural mechanisms for anhedonia with different levels.

The current study confirmed a correlation between RN connectivity and depressive anhedonia. The findings suggested that in the total sample, there was a consistent and negative correlation between anhedonia and striatum connectivity with the insula, inferior frontal cortex, AC, thalamus, and putamen. In the RMD group, there was a different correlation model, i.e., a consistent positive correlation between the severity and frequency of anhedonia and the AC, thalamus, a different negative correlation between the severity of anhedonia and striatum connectivity with the caudate. No significant correlation between anhedonia and striatum connectivity was indicated in the NC and MDD groups, which suggested that too low or too high onprevalence of anhedonia in the subgroup muted the correlation between symptoms of anhedonia and striatum connectivity, while a proportionate occurrence of anhedonia in the RMD group could reflect acorrelation trend. Although anhedonia score goes MDD > RMD > NC, however, as discussed previously, anhedonia is a core symptom of depressive episode, anhedonia itself can not represent a depressive episode [20]. Thus, the correlation between anhedonia score and RN activation can not be equal to the correlation between depressive scores and RN activation: anhedonia and depressive episode share similar but subtly-different RN mechanism (*Hypothesis 1*). Interestingly,in all group analysis, right putamen was negatively correlated with severity of anhedonia, while left putamen was negatively correlated with frequency of anhedonia, which was consistent with previous reporting [46]. Moreover, in RMD group analysis, right caudate negatively correlated with severity of anhedonia and positively correlated with frequency of anhedonia, while this cortex did not report a correlation in all group analysis, which verified recent animal experiment in a sense [47]. The reason might be that the reward network mechanisms underling the severity and frequency of anhedonia are different (hemisphere preference in putamen activation, and direction preference in caudate activation in RMD), since that high frequency (almost every day) may be mild in severity (I don't like things as usual), which needs further evidence. The current study confirmed that RMD patients possess a different RN pattern of anhedonia compared with MDD(*Hypothesis 2*), first report in its kind, as far as we know, which enriches the understanding about the neural mechanism underling anhedonia during the process of depression. Together, the correlation results suggested unique roles ofputamen, insula and inferior frontal cortex in anhedonia, a role of caudate in the remission of depression, and a role of AC and thalamus as trait-like characteristic of anhedonia and depression remission, which were shared by the whole sample and RMD group.

Analysis of the functional connectivity indicated that the MDD group showed weaker connectivity between the striatum seed and the inferior frontal cortex and insula. In contrast, the RMD group possessed stronger connectivity between the inferior frontal cortex and insula and the striatum seed. The findings suggested that weaker connectivity between the striatum seed and the inferior frontal cortex and insula contributes to episodes of depression, while the remission of depression compensates for the decreased functional connectivity. The results indicate a potential insula and inferior frontal mechanism in depressiveepisode, i.e., decreased connectivity between the striatum and insula and inferior frontal cortex might result in an episode of depression [48–50], which broadens our understanding of the RN mechanism in depressive episode.

One improvement of the current study over the previous literature was that this study observed the RN mechanism in remitted depression. The functional connectivity results confirmed that the RMD group exhibited weaker connectivity between the striatum seed and caudate compared with patients with current depression and the normal controls (*Hypothesis 2*).The results suggest that the remitted patients might maintain an emotional balance through decreased

connectivity between the striatum seed and caudate, which reveals a potential caudate mechanism in the recovery of depression, i.e., individuals with remitted depression might reach an emotional balance via decreased connectivity between the ventral striatum and caudate [51].

Importantly, both patients with current and remitted depression possessed weaker connectivity between the striatum seed and AC, while MDD group indicated higher ALFF amplitude in AC compared with RMD group. The results indicated that the weaker connectivity between the striatum seed and the AC remained unchanging before and after the remission of depression, which might be a trait-like biomarker of depression constantly correlated with the occurrence or recurrence of depression. Combined with task-state fMRI in our previous reporting [52], i.e., MDD group showed higher activation in AC during a cue-target task compared with RMD group, the results suggest a potential antidepressant target in clinic, i.e., the AC might be a candidate cortex in the prevention and treatment of depression.

T1 scan data was further analyzed and indicated that MDD patients had bigger volume in thalamus compared with the RMD group. Combining the T1 result with correlation findings and the functional connectivity results, our data suggested that the insula and inferior frontal cortex characterized by anhedonia and depressive episode, the caudate was more involved in the remission of depression, the putamen was involved only in the depressive anhedonia, and the AC and thalamus represented a trait-like mechanism in anhedonia, depressive episode and remission. The results confirm that anhedonia is a core symptom of depressive episode, which might be a core pathological mechanism underlying episode of depression [16,53], i.e., there areshared AC, thalamus, insula, and inferior frontal mechanisms between depressive episode and anhedonia, however, anhedonia has unique putamen mechanism (**Hypothesis 1**).

Limitations: First, the sample size was small in this study, which affectedthe explanatory power of ourfindings. Second, the brain volume or intelligence level of subjects were not evaluated,which is common in the research using fMRI [54]. Third, the anhedonia score was evaluated by two items of scales (The first item of PHQ and the fourth item of BDI-II) instead of more detailed instrument, and the neural activation of anhedonia was observed only by resting-state data which could be well observed in resting-state neural image [12, 15]. Finally, patients with current episode were all medicated, which has been confirmed thatthere was no obvious effect of medicine on cerebral connectivity [55,56].

Notably, this study included patients with current and remitted depression and confirmed that MDD and RMD patients possess different RN pattern, as well as depressive episode and anhedonia.The findings help to suggest candidate cortex in the treatment of depression. Specifically, caudate involves more in remitted depression, which might be an indicator of remitted stage, and be a potential target in future antidepressant treatment such as deep brain stimulation or medication. Putamen indicatesonly anhedonia but not depressive episode, which might be used to discriminate anhedonia and depressive episode. Insula and inferior frontal involve in both depressive episode and anhedonia, which might be cortex markers of both situations. Notably, AC and thalamusare trait-like markers of bothcurrent and remitted depression, which might involve in different stages of depressive progress, and reflected trait-like neural changes in depression. Future neural experiments including both MDD and RMD are warranted to confirm these findings.

## Conclusions

The current study is among the first to confirm that RMD patients possess different RN pattern of anhedonia compared with MDD. The findings that the AC, caudate, insula, and inferior frontal cortexes function differently in anhedonia, depressive episodes, and depression remission, provide important suggestions for the therapy of clinical depression, *i.e.,* the caudate is more involved in the recoveryfrom depression, the putamen is involved more in anhedonia, the insula and inferior frontal cortex represent depressive episodes and anhedonia, and the AC and thalamus represent anhedonia, depressive episodes, and depression remission. Hence, the findings offerreliable evidence for the caudate mechanism in remitted depression, the insula and inferior frontal mechanisms in depressive episodes/anhedonia, and the trait-like AC and thalamus mechanisms in both anhedonia/depressive episodes and remission, which suggest candidate biomarkers for the treatment of clinical depression.

## Supporting information

**S1 Fig. Flow chart of participant recruitment.**
(TIF)

**S2 Fig. The reward network (RN, ventral striatum seed, 21, 9, 0).**
(TIF)

**S3 Fig. Scatterplots of correlation between functional connectivity and depressive anhedonia.**
(TIF)

**S4 Fig. Group difference on function connectivity with the whole brain (striatum seed).**
(TIF)

**S5 Fig. Bigger thalamus volume in MDD compared with the RMD group.**
(TIFF)

**S1 Data. RN group data.**
(XLSX)

## Author contributions

**Data curation:** Xiaoxiao Sun, Zhaohua Chen.

**Formal analysis:** Xiaoxiao Sun, Chenxuan Jin, Chun Cui.

**Funding acquisition:** Qin Dai.

**Investigation:** Xiaoxiao Sun.

**Writing – original draft:** Chenxuan Jin.

**Writing – review & editing:** Qin Dai.

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
