## [Decision Letter · Decision Letter 0]

9 Jul 2025

PONE-D-25-24743
Reward network mechanism in anhedonia and depression
PLOS ONE

Dear Dr. Dai,

Thank you for submitting your manuscript to PLOS ONE. After careful consideration, we feel that it has merit but does not fully meet PLOS ONE’s publication criteria as it currently stands. Therefore, we invite you to submit a revised version of the manuscript that addresses the points raised during the review process.

We look forward to receiving your revised manuscript.

Kind regards,

Rosemary Bassey, Ph.D.

Academic Editor

PLOS ONE

Journal Requirements:

2. Please describe in your methods section how capacity to provide consent was determined for the participants in this study. Please also state whether your ethics committee or IRB approved this consent procedure. If you did not assess capacity to consent please briefly outline why this was not necessary in this case.

3. In the online submission form, you indicated that The data that support the findings of this study are available from the corresponding author upon reasonable request.].

4. Thank you for stating the following financial disclosure: [Doctor Dai claimed that this work was supported by the key project of natural science foundation of Chongqing (cstc2020jcyj-zdxmX0009), the Key project and innovation project of People's Liberation Army of China (2021HL003).]. 

5. Thank you for stating the following in the Competing Interests section: [Authors claimed receiving research funding from key project of nature science foundation of Chongqing (cstc2020jcyj-zdxmX0009), the Key project and innovation project of People's Liberation Army of China (2021HL003).Other authors report no biomedical financial interests or potential conflicts of interest.

All authors state that the content has not been published or submitted for publication elsewhere.].

6. Thank you for stating the following in the Acknowledgments Section of your manuscript: [Authorsclaimed that this study was supported by key project of nature science foundation of Chongqing (cstc2020jcyj-zdxmX0009), the Key project and innovation project of People's Liberation Army of China (2021HL003). The authors thank clinical doctors who devote to patients recruitment, including Dr. Dr. L. D. Hu from Geleshan Psychiatry Hospital, Dr. X. Y. Zhou from Psychiatry center of Chongqing, Dr.Y.He from 324 Hospital of Chong Qing. We appreciate the endeavor of all graduate students who participated this study as research assistants. We also thank all subjects who took part in this study, especially the clinical patients.]

Please remove any funding-related text from the manuscript and let us know how you would like to update your Funding Statement. Currently, your Funding Statement reads as follows: [Doctor Dai claimed that this work was supported by the key project of natural science foundation of Chongqing (cstc2020jcyj-zdxmX0009), the Key project and innovation project of People's Liberation Army of China (2021HL003).]. 

7. Your ethics statement should only appear in the Methods section of your manuscript. If your ethics statement is written in any section besides the Methods, please move it to the Methods section and delete it from any other section. Please ensure that your ethics statement is included in your manuscript, as the ethics statement entered into the online submission form will not be published alongside your manuscript.

8. We notice that your supplementary figures are uploaded with the file type 'Figure'. Please amend the file type to 'Supporting Information'. Please ensure that each Supporting Information file has a legend listed in the manuscript after the references list.

9. We notice that your supplementary figures are included in the manuscript file. Please remove them and upload them with the file type 'Supporting Information'. Please ensure that each Supporting Information file has a legend listed in the manuscript after the references list.

Reviewers' comments:

Reviewer's Responses to Questions

**Comments to the Author**

1. Is the manuscript technically sound, and do the data support the conclusions?

Reviewer #1: Yes

Reviewer #2: Yes

Reviewer #3: Yes

2. Has the statistical analysis been performed appropriately and rigorously? 

Reviewer #1: Yes

Reviewer #2: Yes

Reviewer #3: Yes

3. Have the authors made all data underlying the findings in their manuscript fully available?

Reviewer #1: Yes

Reviewer #2: Yes

Reviewer #3: Yes

4. Is the manuscript presented in an intelligible fashion and written in standard English?

Reviewer #1: Yes

Reviewer #2: Yes

Reviewer #3: Yes

5. Review Comments to the Author

Reviewer #1: The manuscript by Sun D.X. and co-authors analyses the clinical data on anhedonia and depression from the side of reward network mechanism.

The manuscript is well-prepared, gives enough data to confirm the author's statements and claims.

With this in mind, presented manuscript is a good work that may be published in PLOS One journal.

Per contrary, it has several important points I would like to address to the authors and I expect they will correct them:

1. Please discuss the recent data on brain structures (that you study in your work) interactions in regulation of anhedonia and depression. I suppose there are a lot of new experimental articles that show how these brain regions interact with each other in controlling behavior employing cell and tissue histology and physiology methods.

2. The references list is very old. Two articles published on 2020 and 2021 are the newest ones in this manuscript. The authors should update the references list by adding the newer publications.

3. In addition to the study limitations, the authors should also include the perspectives subsection, where they should write in details on how potentially this work and the data may be used in further research.

4. Sometimes the English usage appears strange, please check the spelling.

Reviewer #2: This study is an interesting research topic that has demonstrated the different reward network involvement of anhedonia and depression. However, in order to increase readers' understanding, it seems necessary to revise the following parts.

1. Throughout the manuscript, certain words (aka. correlate) are repeated too much, and the fact that it is unnecessarily the first study is used repeatedly for discussion. English correction is needed throughout the paper.

2. With this study alone, it seems unreasonable for the reward network to evaluate which of the state or trace markers will be suitable. Nevertheless, if the authors describe it as suitable as a trace marker, the evidence for this should be described.

3. In the introduction, authors argued that anhedonia is a core symptom of depression, and MDD and anhedonia may have different natural mechanisms. Could you explain this issue more detail in evidence?

4. pp. 42-44. When describing the conflicting results of previous studies on decreased depression, hyperactivation was described before without mentioning a specific brain region, and hypoactivation was described later in putamen and caudate. The authors should specify the specific brain region that is hyperactivated.

5. In the last part of the introduction, the authors established two hypotheses, talking about the purpose of the study. The authors should further describe the scientific basis for this hypothesis.

6. In Figures 1-5, specific positions with the annotation (A,B,C) should be established and specify each figure location in the manuscript.

Reviewer #3: This is an observational study with a cross-sectional design that employed resting-state functional magnetic resonance imaging (rs-fMRI) to investigate patterns of functional brain connectivity. The primary objective was to compare connectivity differences across three groups: individuals with remitted depression, individuals currently experiencing a depressive episode, and individuals with no history of psychiatric disorders. The methodology was clearly described, utilizing standardized and validated instruments appropriate for the target population and suitable for the collection and analysis of neurofunctional data.

1. Is the manuscript technically sound, and do the data support the conclusions?

Although the sample size is relatively small, the study presents valuable contributions to the theoretical understanding of depression, as well as to the development of clinical management strategies. It highlights promising directions for relapse prevention and the personalization of treatment approaches.

2. Has the statistical analysis been performed appropriately and rigorously?

Yes. The statistical analysis was appropriately conducted and allowed for the exploration of the data in line with the study’s objectives.

3. Have the authors made all data underlying the findings in their manuscript fully available?

Yes, the authors have made all relevant data available.

4. Is the manuscript presented in an intelligible fashion and written in standard English?

Yes, the manuscript is clearly written and presented in standard academic English.

5. Additional Comments to the Author

The topic addressed in this manuscript is highly relevant in the fields of neuroscience and psychiatry, offering important insights into the neurobiological mechanisms underlying depression. Notably, the identification of alterations in functional brain connectivity across different phases of the disorder contributes to the growing understanding of its pathophysiology.

A key strength of the study lies in its potential to identify neural biomarkers of depression. The detection of distinct connectivity patterns among the groups studied may support the development of diagnostic and prognostic markers related to the presence, absence, or remission of the disorder. Furthermore, these findings provide a foundation for more individualized therapeutic interventions, such as neuromodulation techniques or psychotherapeutic approaches targeting specific neural circuits.

Recommendation: The manuscript is of scientific merit and relevance. It is therefore recommended for publication.

6. PLOS authors have the option to publish the peer review history of their article (what does this mean?). If published, this will include your full peer review and any attached files.

Reviewer #1: No

Reviewer #2: **Yes: **Jae Yong Choi

Reviewer #3: No

---

## [Author Response · Author response to Decision Letter 1]

25 Aug 2025

Dear Ph.D. Rosemary Bassey,

Thank you for giving us an opportunity to revise the manuscript. Please find enclosed revised version of our manuscript entitled “Reward network mechanism in anhedonia and depression” for your consideration.

Many thanks for your constructive and detailed comments on the manuscript, which substantially improves the paper. In this revision version, we have addressed all of comments made by the reviewers (see below). With these changes, we hope that the manuscript will be of more relevant to the readership of your journal.

This manuscript is not in consideration elsewhere and the author reports no conflicts of interest. Moreover, the co-author has approved the revised version making significant contributions to the writing and conceptualization of the manuscript.

If I can be of any further assistance, please do not hesitate to contact me at my email address (daiqin101@hotmail.com). Thank you for your consideration.

Yours sincerely,

Dai Qin

COMMENTS FOR THE AUTHOR:

Journal Requirements:

Author’s responding: Thank you for the detailed suggestion and revised accordingly.

2. Please describe in your methods section how capacity to provide consent was determined for the participants in this study. Please also state whether your ethics committee or IRB approved this consent procedure. If you did not assess capacity to consent please briefly outline why this was not necessary in this case.

Author’s responding: Thank you for the important suggestion and revised accordingly. The details about how capacity to provide consent in participants were added in Participants (Line 80-82) “…The exclusion criteria were as follows: current or history of any psychopathology and/or drug and alcohol abuse infection, drug dependence, allergies within the last half month, physical trauma, severe physical disease, learning or cognitive disability…” and Procedure (Line 98-101,102-103) “…The study protocol including participants’ capacity to provide consent in accordance with the Declaration of Helsinki was approved by the Ethics Committee of Human Research of the Army Medical University (2020-036-02) and Chinese Clinical Trial registry center (ChiCTR2100044258)…The potential subjects (patients and normal controls) were screened for recruitment and their capacity to provide consent by trained personnel…”

3. In the online submission form, you indicated that The data that support the findings of this study are available from the corresponding author upon reasonable request.].

Author’s responding: Thank you for the important suggestion and revised accordingly. We added the information in Data Availability Statement “Data Availability Statement

The behavioral data that support the findings of this study were included in Supplementary materials. Due to reframing from military funding (2021HL003 and 2023yjsB02), the neural data could not be placed publicly and was available from the corresponding author upon reasonable request. To ensure the long term stability and availability of data, the neural data was packed and stored in three copies by a non-author institution (Military Psychology Committee institute, Chinese Society of Social Psychology) to ensure persistent or long-term data storage and availability, which was available from the official email address of this committee (jsxlxzwh@163.com)” and Cover letter.

4. Thank you for stating the following financial disclosure: [Doctor Dai claimed that this work was supported by the key project of natural science foundation of Chongqing (cstc2020jcyj-zdxmX0009), the Key project and innovation project of People's Liberation Army of China (2021HL003).].

Author’s responding: Thank you for the important suggestion and revised accordingly. See details in Funding Declaration “Doctor Dai claimed that this work was supported by Social Science Foundation of Chongqing (2023NDYB94), the Key project and innovation project of People's Liberation Army of China (2021HL003), and the education project of Army medical university (2023yjsB02). The funders had no role in study design, data collection and analysis, decision to publish, or preparation of the manuscript, only reframed the sharing of original neural data.” and Cover letter.

5. Thank you for stating the following in the Competing Interests section: [Authors claimed receiving research funding from key project of nature science foundation of Chongqing (cstc2020jcyj-zdxmX0009), the Key project and innovation project of People's Liberation Army of China (2021HL003).Other authors report no biomedical financial interests or potential conflicts of interest.

All authors state that the content has not been published or submitted for publication elsewhere.].

Author’s responding: Thank you for the important suggestion and revised accordingly. See details in Declaration of interest statement “Authors claimed receiving research funding from Social Science Foundation of Chongqing (2023NDYB94), the Key project and innovation project of People's Liberation Army of China (2021HL003), and the education project of Army medical university (2023yjsB02). Other authors report no biomedical financial interests or potential conflicts of interest. This does not alter our adherence to PLOS ONE policies on sharing behavioral data, only limits on original neural data sharing from military funding (2021HL003 and 2023yjsB02). All authors state that the content has not been published or submitted for publication elsewhere.” and Cover letter.

6. Thank you for stating the following in the Acknowledgments Section of your manuscript: [Authorsclaimed that this study was supported by key project of nature science foundation of Chongqing (cstc2020jcyj-zdxmX0009), the Key project and innovation project of People's Liberation Army of China (2021HL003). The authors thank clinical doctors who devote to patients recruitment, including Dr. Dr. L. D. Hu from Geleshan Psychiatry Hospital, Dr. X. Y. Zhou from Psychiatry center of Chongqing, Dr.Y.He from 324 Hospital of Chong Qing. We appreciate the endeavor of all graduate students who participated this study as research assistants. We also thank all subjects who took part in this study, especially the clinical patients.]

Please remove any funding-related text from the manuscript and let us know how you would like to update your Funding Statement. Currently, your Funding Statement reads as follows: [Doctor Dai claimed that this work was supported by the key project of natural science foundation of Chongqing (cstc2020jcyj-zdxmX0009), the Key project and innovation project of People's Liberation Army of China (2021HL003).].

Author’s responding: Thank you for the important suggestion and revised accordingly. See details in Acknowledgement “The authors thank clinical doctors who devote to patients recruitment, including Dr. Dr. L. D. Hu from Geleshan Psychiatry Hospital, Dr. X. Y. Zhou from Psychiatry center of Chongqing, Dr.Y.He from 324 Hospital of Chong Qing. We appreciate the endeavor of all graduate students who participated this study as research assistants. We also thank all subjects who took part in this study, especially the clinical patients.” and Cover letter.

7. Your ethics statement should only appear in the Methods section of your manuscript. If your ethics statement is written in any section besides the Methods, please move it to the Methods section and delete it from any other section. Please ensure that your ethics statement is included in your manuscript, as the ethics statement entered into the online submission form will not be published alongside your manuscript.

Author’s responding: Thank you for the valuable suggestion and revised accordingly. Ethic statement were moved to Procedures (Line 96-101): “All procedures performed in studies involving human participants were in accordance with the ethical standards of the institutional and/or national research committee and with the 1964 Helsinki Declaration and its later amendments or comparable ethical standards. The study protocol including participants’ capacity to provide consent in accordance with the Declaration of Helsinki was approved by the Ethics Committee of Human Research of the Army Medical University (2020-036-02) and Chinese Clinical Trial registry center (ChiCTR2100044258)…”

8. We notice that your supplementary figures are uploaded with the file type 'Figure'. Please amend the file type to 'Supporting Information'. Please ensure that each Supporting Information file has a legend listed in the manuscript after the references list.

Author’s responding: Thank you for the important suggestion and revised accordingly. All supplementary Figures were uploaded as 'Supporting Information'.

9. We notice that your supplementary figures are included in the manuscript file. Please remove them and upload them with the file type 'Supporting Information'. Please ensure that each Supporting Information file has a legend listed in the manuscript after the references list.

Author’s responding: Thank you for the important suggestion and revised accordingly. All supplementary Figures had a legend listed in the manuscript after the references list and were uploaded as 'Supporting Information'.

Author’s responding: Thank you for the important suggestion and revised accordingly. We checked all references and made minor revisions in volume and pages in some of references (11, 12, 45), and confirmed that the references were complete and correct in the current version.

Reviewer #1:

The manuscript by Sun D.X. and co-authors analyses the clinical data on anhedonia and depression from the side of reward network mechanism.

The manuscript is well-prepared, gives enough data to confirm the author's statements and claims.

With this in mind, presented manuscript is a good work that may be published in PLOS One journal.

Author’s responding: Many thanks for your constructive and detailed comments on the manuscript, which substantially improves the paper.

Per contrary, it has several important points I would like to address to the authors and I expect they will correct them:

1. Please discuss the recent data on brain structures (that you study in your work) interactions in regulation of anhedonia and depression. I suppose there are a lot of new experimental articles that show how these brain regions interact with each other in controlling behavior employing cell and tissue histology and physiology methods.

Author’s responding: Thank you for the valuable suggestion and revised accordingly. We described latest reporting about neural mechanism in regulation of anhedonia and depression in 1st and 2nd paragraph of Introduction (Line 12-14, 29-32): “…Blunted left ventral striatum response to reward was associated with a lifetime depression diagnosis [8]. Recent review indicated that compared with healthy controls, patients with MDD exhibited common activity decreases in the right striatum (putamen, caudate) and subgenual ACC [9]…. Change in anhedonia scores negatively correlated with rsFC after antidepressant treatment [17]. A previous study further reported that anhedonia symptom and depressive mood were associated with similar striatal circuits in MDD patients with money-reward task [16]. Indeed, RN was indicated associated with both anhedonia and depression [18]…”

2. The references list is very old. Two articles published on 2020 and 2021 are the newest ones in this manuscript. The authors should update the references list by adding the newer publications.

Author’s responding: Thank you for the important suggestion and revised accordingly. We added 14 recent literatures in this version (literature 6, 8, 9, 17, 18, 19, 21-22, 24-26, 35, 46, 47). See references for details.

3. In addition to the study limitations, the authors should also include the perspectives subsection, where they should write in details on how potentially this work and the data may be used in further research.

Author’s responding: Thank you for the inspiring suggestion and revised accordingly. A separate paragraph about the perspective of current findings were added after Limitation and before Conclusion (Line 293-302): “Notably, this study included patients with current and remitted depression and confirmed that MDD and RMD patients possess different RN pattern, as well as depressive episode and anhedonia. The findings help to suggest candidate cortex in the treatment of depression. Specifically, caudate involves more in remitted depression, which might be an indicator of remitted stage, and be a potential target in future antidepressant treatment such as deep brain stimulation or medication. Putamen indicates only anhedonia but not depressive episode, which might be used to discriminate anhedonia and depressive episode. Insula and inferior frontal involve in both depressive episode and anhedonia, which might be cortex markers of both situations. Notably, AC and thalamus are trait-like markers of both current and remitted depression, which might involve in different stages of depressive progress, and reflected trait-like neura

---

## [Decision Letter · Decision Letter 1]

4 Sep 2025

Reward network mechanism in anhedonia and depression

PONE-D-25-24743R1

Dear Dr. Qin Dai,

We’re pleased to inform you that your manuscript has been judged scientifically suitable for publication and will be formally accepted for publication once it meets all outstanding technical requirements.

Kind regards,

Rosemary Bassey, Ph.D.

Academic Editor

PLOS ONE

Additional Editor Comments (optional):

Reviewer #1:

Reviewer #2:

Reviewers' comments:

Reviewer's Responses to Questions

**Comments to the Author**

1. If the authors have adequately addressed your comments raised in a previous round of review and you feel that this manuscript is now acceptable for publication, you may indicate that here to bypass the “Comments to the Author” section, enter your conflict of interest statement in the “Confidential to Editor” section, and submit your "Accept" recommendation.

Reviewer #1: All comments have been addressed

Reviewer #2: All comments have been addressed

2. Is the manuscript technically sound, and do the data support the conclusions?

Reviewer #1: Yes

Reviewer #2: Yes

3. Has the statistical analysis been performed appropriately and rigorously? 

Reviewer #1: Yes

Reviewer #2: Yes

4. Have the authors made all data underlying the findings in their manuscript fully available?

Reviewer #1: Yes

Reviewer #2: No

5. Is the manuscript presented in an intelligible fashion and written in standard English?

Reviewer #1: Yes

Reviewer #2: Yes

6. Review Comments to the Author

Reviewer #1: The authors have cleared all the points and answered all my questions. The manuscript may now be accepted for publication.

Reviewer #2: Thank you very much. All comments made by reviewers were properly reflected in the revised manuscript.

7. PLOS authors have the option to publish the peer review history of their article (what does this mean?). If published, this will include your full peer review and any attached files.

Reviewer #1: No

Reviewer #2: **Yes: **Jae Yong Choi

---

## [Editor Report · Acceptance letter]

PONE-D-25-24743R1

PLOS ONE

Dear Dr. Dai,

I'm pleased to inform you that your manuscript has been deemed suitable for publication in PLOS ONE. Congratulations! Your manuscript is now being handed over to our production team.

Kind regards,

on behalf of

Dr. Rosemary Bassey

Academic Editor

PLOS ONE